# Photocatalytic Activity of Silver-Based Biomimetics Composites

**DOI:** 10.3390/biomimetics6010004

**Published:** 2021-01-04

**Authors:** Abniel Machín, Loraine Soto-Vázquez, Carla Colón-Cruz, Carlos A. Valentín-Cruz, Gerardo J. Claudio-Serrano, Kenneth Fontánez, Edgard Resto, Florian I. Petrescu, Carmen Morant, Francisco Márquez

**Affiliations:** 1Arecibo Observatory, Universidad Ana G. Méndez-Cupey Campus, San Juan 00926, Puerto Rico; 2Materials Characterization Center, Molecular Sciences Research Center, University of Puerto Rico, San Juan 00926, Puerto Rico; lorainesoto@gmail.com (L.S.-V.); edgard.resto@upr.edu (E.R.); 3Nanomaterials Research Group, School of Natural Sciences and Technology, Universidad Ana G. Méndez-Gurabo Campus, Gurabo 00778, Puerto Rico; ccolon265@email.uagm.edu (C.C.-C.); cvalentin51@email.uagm.edu (C.A.V.-C.); gclaudio17@email.uagm.edu (G.J.C.-S.); 4Department of Chemistry, University of Puerto Rico-Río Piedras, San Juan 00925, Puerto Rico; kenneth.fontanez@gmail.com; 5IFToMM-ARoTMM, Bucharest Polytechnic University, 060042 Bucharest, Romania; fitpetrescu@gmail.com; 6Department of Applied Physics, Autonomous University of Madrid and Instituto de Ciencia de Materiales Nicolas Cabrera, 28049 Madrid, Spain; c.morant@uam.es

**Keywords:** photocatalysis, hydrogen production, ciprofloxacin, silver nanoparticles, TiO_2_, ZnO

## Abstract

Different Ag@TiO_2_ and Ag@ZnO catalysts, with nanowire (NW) structure, were synthesized containing different amounts of silver loading (1, 3, 5, and 10 wt.%) and characterized by FE-SEM, HRTEM, BET, XRD, Raman, XPS, and UV–vis. The photocatalytic activity of the composites was studied by the production of hydrogen via water splitting under UV–vis light and the degradation of the antibiotic ciprofloxacin. The maximum hydrogen production of all the silver-based catalysts was obtained with a silver loading of 10 wt.% under irradiation at 500 nm. Moreover, 10%Ag@TiO_2_ NWs was the catalyst with the highest activity in the hydrogen production reaction (1119 µmol/hg), being 18 times greater than the amount obtained with the pristine TiO_2_ NW catalyst. The most dramatic difference in hydrogen production was obtained with 10%Ag@TiO_2_-P25, 635 µmol/hg, being 36 times greater than the amount reported for the unmodified TiO_2_-P25 (18 µmol/hg). The enhancement of the catalytic activity is attributed to a synergism between the silver nanoparticles incorporated and the high surface area of the composites. In the case of the degradation of ciprofloxacin, all the silver-based catalysts degraded more than 70% of the antibiotic in 60 min. The catalyst that exhibited the best result was 3%Ag@ZnO commercial, with 99.72% of degradation. The control experiments and stability tests showed that photocatalysis was the route of degradation and the selected silver-based catalysts were stable after seven cycles, with less than 1% loss of efficiency per cycle. These results suggest that the catalysts could be employed in additional cycles without the need to be resynthesized, thus reducing remediation costs.

## 1. Introduction

During the last decade, there has been a significant increase in discussions related to the production of energy from renewable sources [1,2]. This is mainly due to the increase in global temperatures and climate change, caused by the burning of fossil fuels and release of greenhouse gases such as carbon dioxide (CO_2_) [2]. For this reason, countless efforts around the world have focused on developing technologies that allow the production of energy from renewable sources in a cost-effective way. Within these efforts, inspiration and answers are sought in nature. Materials that can mimic what nature has perfected over thousands or millions of years are investigated in the hope that they can be manufactured, used, developed, and employed upscale.

Developing active and robust water oxidation catalysts is the key to constructing efficient artificial photosynthetic systems [3,4,5,6]. The oxidation of water is the primary and key reaction of overall water splitting, but it requires the transfer of 4 electrons (e^−^) and 4H^+^ with continuous formation of O-O bonds, leading to high energy barriers and slow kinetics [6]. Therefore, great efforts have been devoted to developing efficient catalysts, including homogeneous, heterogeneous, and especially biomimetic catalysts, to facilitate the oxygen evolution reaction (OER) in the past few decades [6].

In terms of biomimetics catalysts, water oxidation occurs in nature at the so-called CaMn_4_O_5_ oxygen evolution complexes surrounded by many amino acid residues in photosystem (II) (PSII) [5,6]. To date, water oxidation catalysts have been integrated into photocatalytic and photoelectrocatalytic (PEC) systems to construct artificial photosynthetic systems [4,5,6]. Selective deposition on special sites of light-harvesting semiconductors and interface engineering with charge transfers are some of the modifications that have been implemented to improve the efficiency of photosystems [6].

Semiconductors such as TiO_2_ and ZnO have been extensively used, mainly due to their nontoxicity, low cost, and high chemical stability [7,8,9,10]. One of the main disadvantages of using TiO_2_ and ZnO as photocatalysts is their wide band gap energy (i.e., 3.2 eV for TiO_2_ anatase; 3.37 eV for ZnO wurtzite). With these energy gaps, only UV light can be used [8,9]. This fact must be taken into consideration since only 4% of the total radiation that comes from the sun is in the UV region, whereas 50% is in the visible region of the electromagnetic spectrum [10]. It is for this reason that recent investigations have focused on modifying the band gap of the catalysts. Some of the chemical modifications include the doping of TiO_2_ and ZnO with metal and non-metal elements [7,8,9,10]. The incorporation of noble metals, such as silver (Ag), on the surfaces of these semiconductors has gained substantial interest in recent years due to the ability of the noble metal nanoparticles to reduce the fast recombination of the photogenerated charge carriers, enabling the use of visible light [11,12,13,14,15,16,17,18]. By reducing the photogenerated charge carriers, the UV activity is increased due to the electron transfer from the conduction band (CB) of TiO_2_ or ZnO to the noble metal nanoparticles [12]. The photoactivity in the visible spectral range can be explained due to the surface plasmon resonance effect and charge separation by the transfer of photoexcited electrons from the metal nanoparticles to the CB of TiO_2_ or ZnO [11,12,13,14,15,16,17,18].

The development of silver-based catalysts (Ag@TiO_2_ and Ag@ZnO) for hydrogen production via water splitting and the degradation of organic pollutants have been attracting attention in recent years. For example, Saravanan et al. [12] synthesized Ag@TiO_2_ photocatalysts for the production of hydrogen and degradation of methyl orange dye (MO). The catalysts exhibited good visible light activity and excellent stability over three cycles for the aqueous phase photocatalytic degradation of MO (38 μmol/h g) and excellent hydrogen production from water splitting (910 μmol/h g). The results were justified by the incorporation of Ag onto TiO_2_, which improved the photophysical properties, narrowing the band gap and suppressing the charge carrier recombination. On the other hand, Mao et al. [14] synthesized Ag@TiO_2_ with different structures (nanoparticles, nanoflakes, and nanorods). The as-synthesized Ag@TiO_2_ compounds were employed as catalysts for the photodegradation of MO under visible light irradiation, showing that Ag@TiO_2_ compounds display a catalytic performance superior to pure TiO_2_. The catalyst with 20 wt.% Ag exhibited the best photocatalytic activity after 120 min of treatment, with photodegradations of ca. 98.9% for MO under visible light irradiation. This was explained by the fact that a suitable amount of Ag could restrain the recombination of the photoproduced charges and extend the range of the light absorption. Meanwhile, Mou et al. [15] developed three-dimensional (3D) Ag@ZnO assemblies with porous nanosheets as building blocks to produce hydrogen via water splitting and the degradation of 4-nitrophenol, demonstrating superior sunlight-driven photocatalytic activities, with nearly 100% degradation of 10 ppm 4-nitrophenol and 0.4436 mmol g^−1^ h^−1^ of hydrogen production. The authors attribute these results to the synergistic effect between 2D ZnO porous single crystalline nanosheets and silver, which could not only accelerate the separation and migration efficiency of charge carriers but also enhance the charge collection efficiency. Other authors [16] synthesized Ag@ZnO composites for the degradation of rhodamine B and the production of hydrogen by water splitting. Under ultraviolet, visible-light, simulated sunlight, and microwave-assisted irradiation, the composite exhibited better photocatalytic properties for the photocatalytic degradation of rhodamine B compared to that of P25 and ZnO. Moreover, the composite was investigated as a catalyst for the degradation of four dyes with different structures under UV conditions, exhibiting good degradation performance. In addition, the photocatalytic hydrogen production experiments showed that Ag@ZnO had some ability to produce photocatalytic hydrogen.

Ciprofloxacin is a fluoroquinolone which constitutes one of the major families of antibiotics [19]. Ciprofloxacin is employed for treating infectious diseases and is effective against both Gram-positive and Gram-negative bacteria. During their use in humans, these antibiotics are incompletely metabolized and are excreted mostly through urine and stool. Ciprofloxacin has been detected in appreciable quantities in continental waters and reservoirs, as it is a compound which is highly resistant to degradation. Antibiotics, and particularly ciprofloxacin, enter the body through drinking water, allowing bacteria to become resistant to these compounds. For these reasons, the purposes of this investigation were twofold: (1) to synthesize different weight percentages (1, 3, 5, and 10 wt.%) of silver-based catalysts (Ag@TiO_2_ and Ag@ZnO); and (2) to study the photocatalytic activity of the as-synthesized composites in two processes of current relevance—the production of hydrogen via water splitting and the photodegradation of ciprofloxacin.

## 2. Materials and Methods

### 2.1. Reagents

All reagents were used as received and all the experimental solutions were prepared using ultra-pure water (Milli-Q water, 18.2 MΩcm^−1^ at 25 °C). Acetone, isopropyl alcohol +99.9%, and HCl 37% (ACS Reagent) were provided by Acros Chemicals (Newark, NJ, USA). TiCl_4_ 99.9% was obtained from Fisher Scientific (Pittsburgh, PA, USA). Degussa P25 (Degussa, nanopowder with 21-nm particle size, 35–65 m^2^·g^−1^ surface area, +99.5%), AgNO_3_ (99.99+%, trace metal basis), Zn(C_2_H_3_O_2_)_2_ (98+%, ACS Reagent), and NaBH_4_ +99.9% were provided by Sigma Aldrich (Milwaukee, WI USA). The commercial zinc oxide (99.99%) was obtained from Alfa Aesar (Ward Hill, MA, USA). Si <100> substrates, p-type boron-doped, provided by El-CAT (Ridgefield Park, NJ, USA), were used as substrates for the hydrothermal growth of TiO_2_ nanowires (TiO_2_ NWs). Finally, for the photocatalytic experiments, ciprofloxacin (C_17_H_18_FN_3_O_3_, ≥98% HPLC), Na_2_S 99.9%, and Na_2_SO_3_ + 98% were also obtained from Sigma Aldrich (Milwaukee, WI, USA).

### 2.2. Synthesis of TiO_2_ and ZnO Nanowires

The synthesis of TiO_2_ NWs was carried out following the method described by Cotto et al. (2013) [20]. In a typical synthesis, 50 mL of concentrated hydrochloric acid and 50 mL of deionized water were mixed in a 200 mL Erlenmeyer flask. After this, 3 mL of the titanium precursor was added by dripping, under agitation, at room temperature. The mixture was magnetically stirred until all solid particles were dissolved (approximately 10 min). Then, the solution was placed into 30-mL Teflon-lined stainless-steel autoclaves. Next, single crystal silicon substrates, Si <100>, were introduced inside the autoclaves. Autoclaves were maintained at 180 °C for 24 h. The resulting TiO_2_ NWs, grown on the surfaces of the silicon wafers, were thoroughly washed with deionized water and dried overnight at 60 °C.

In the case of the ZnO NWs, the synthesis consisted of the thermal decomposition of zinc acetate, according to the procedure used by Lin and Li (2009) [21]. For this, 0.5 g of zinc acetate dihydrate was heat-treated at 300 °C for 3 h in an alumina crucible. After the reaction time, the crucible was left to cool down and the product was then collected and sealed at room temperature.

### 2.3. Incorporation of Silver Nanoparticles (Ag NPs)

The incorporation of Ag NPs onto the surfaces of TiO_2_ NWs, TiO_2_-P25, ZnO NWs, and ZnO commercial catalysts consisted of a modification of the method described by Naldoni et al. [22]. A typical synthesis consisted of dispersing 500 mg of the desired catalyst in 40 mL of H_2_O and subsequent treatment with ultrasound for 20 min. After this, the desired amount of the silver precursor was added to the reaction mixture and stirred for 30 min. Finally, a NaBH_4_ solution (10 mg in 10 mL of H_2_O) was added dropwise, under stirring, and allowed to react for 10 min at room temperature. The reaction product was centrifuged, washed three times with deionized water, and dried overnight at 60°C. The different silver-based composites were identified as x%Ag@TiO_2_ NWs, x%Ag@TiO_2_-P25, x%Ag@ZnO NWs, x%Ag@ZnO commercial. The numbers (x%) correspond to the weight percentage of Ag NPs in the sample.

### 2.4. Characterization of the Catalysts

The catalysts were fully characterized by electron microscopy, using field-emission scanning electron microscopy (FE-SEM, Philips XL30 S-FEG; Chatsworth, CA, USA) and high-resolution transmission electron microscopy (HRTEM, JEOL 3000F; Peabody, MA, USA). XPS measurements were obtained on an ESCALAB 220i-XL spectrometer (Waltham, MA, USA), by using the non-monochromated Mg Kα (1253.6 eV) radiation of a twin-anode, operating at 20 mA and 12 kV in the constant analyzer energy mode, with a PE of 50 eV. Brunauer–Emmett–Teller (BET) specific surface areas were measured using a Micromeritics ASAP 2020 (Norcross, GA, USA), according to N_2_ adsorption isotherms, at 77 K. Raman measurements were acquired using a DXR Thermo Raman Microscope (Waltham, MA, USA), employing a 532-nm laser source at 5-mW power and a nominal resolution of 5 cm^−1^. X-ray diffraction (XRD) patterns were obtained in theta/2theta configuration in the range of 20–80° at 2° min^−1^, using a Bruker D8 Advance X-Ray Diffractometer (Billerica, MA, USA), operating at 40 kV and 40 mA. The UV–vis spectra were measured using a Shimadzu UV-1800PC spectrophotometer (Missouri City, TX, USA).

### 2.5. Photocatalytic Experiments

The hydrogen production via water splitting experiments consisted of the following: 100 mL of deionized water was added to a 200 mL quartz reactor flask. Then, 50 mg of the silver-based catalyst was added to the flask containing the water. Solutions of 0.5 M Na_2_S and 0.03 M Na_2_SO_3_ were added as sacrificial reagents to promote photocatalytic activity and increase H_2_ production, due to their action as hole scavengers [23,24]. The reaction mixture was thermostatized at 20 °C, magnetically stirred at 20 rpm, and purged with nitrogen for at least 20 min. Then, the solution was exposed to UV–vis radiation for 120 min using different filters to select the appropriate irradiation wavelength. The produced hydrogen was collected, using nitrogen as gas carrier, and was identified and quantified by Gas Chromatography with a Thermal Conductivity Detector (GC-TCD, Perkin-Elmer Clarus 600; Shelton, CT, USA).

For the photocatalytic degradation experiments, a solar simulator composed of two annular white bulb lights with a total irradiation power of 60 watts was used. A solution of 1 × 10^−5^ M of ciprofloxacin was prepared and then the desired catalyst was added. After adding the catalyst and adjusting the pH to 7, the system was kept in the dark for 30 min to let the system reach adsorption–desorption equilibrium. Then, a small amount of H_2_O_2_ (3 mL, 0.005%) was added; additional oxygen was provided to the system by constant air-bubbling. The white bulb lights were switched on, and the photocatalytic system was maintained under constant stirring and irradiation. The reaction was monitored for a period of 60 min, and the kinetics of photodegradation was studied by taking aliquots every 10 min. After filtering the aliquots with 0.45-μm membrane filters, the samples were analyzed with a Shimadzu UV-1800PC spectrophotometer.

## 3. Results and Discussion

### 3.1. Characterization of Catalysts

Figure 1 shows different FE-SEM images of TiO_2_ NWs and ZnO NWs at different magnifications. The as-synthesized TiO_2_ NWs consist of homogeneous and highly branched structures (Figure 1A). At higher magnifications (Figure 1B), TiO_2_ NWs have a square-like form and are composed of smaller units with the same shape. The specific surface area of the catalyst obtained by BET measurements was 403 m^2^ g^−1^, a large area that may be justified by the highly branched structures and could have relevant effects on the catalytic properties of this material. The FE-SEM images of ZnO catalysts are shown in Figure 1C. ZnO NWs consisted of non-homogenous wires, with differences in size and length. The BET surface area of ZnO NWs (160 m^2^ g^−1^) was clearly less than that observed for TiO_2_ NWs.

After the synthesis of TiO_2_ NWs and ZnO NWs catalysts, different amounts of Ag NPs (1%, 3%, 5%, and 10% wt.%) were deposited onto the surfaces of the as-synthesized catalysts and their commercial forms (TiO_2_-P25 and commercial ZnO) for comparison purposes, using a chemical reduction method. Figure 2 shows the high-resolution transmission electron microscopy (HRTEM) images of the catalysts containing 10 wt.% of Ag. All silver-based catalysts showed a non-homogenous distribution of the metal on the surface, with diameters of ca. < 10 nm (Figure 2). This is a normal characteristic when using sodium borohydride (NaBH_4_) as a reducing agent, and it has been documented in previous works [25,26,27]. The 10%Ag@TiO_2_-P25 (Figure 2A) and 10%ZnO commercial (Figure 2C) composites consisted of supports of different shapes and sizes, of micrometers in length, and both the 10%Ag@TiO_2_ NWs (Figure 2B) and 10%ZnO NWs (Figure 2D) showed significant changes in their morphology compared to the bare catalysts shown in Figure 1. These changes were expected and are also attributed to the synthesis process.

The BET surface areas of the silver-based catalysts with different Ag loadings were analyzed and the results are shown in Table 1. Before the silver deposition, the average surface areas of the as-synthesized TiO_2_ NWs and ZnO NWs were 403 and 160 m^2^·g^−1^, respectively, whereas the surface areas of the commercial ones (TiO_2_-P25 and ZnO) were 53 and 18 m^2^·g^−1^, respectively. These results clearly show that the synthesis processes used to obtain TiO_2_ NWs and ZnO NWs allowed us to obtain high-surface supports. Unexpectedly, the addition of silver nanoparticles on the pristine catalysts resulted in an increase in surface area. This effect is much more significant in composites with the highest percentage of silver on the surface (see Table 1). As was determined by HRTEM, these nanoparticles are dispersed on the surfaces of the catalysts, suggesting that the chemical reduction approach is an efficient method for synthesizing these composites.

The XRD patterns of pristine catalysts and catalysts with 10 wt.% Ag are shown in Figure 3. Figure 3A, corresponding to TiO_2_ NWs (pristine or with 10 wt.% Ag), shows intense peaks at 27° (110), 36° (101), and 55° (211), which have been unambiguously ascribed to the rutile phase (JCPDS 88-1175) [28]. The 10% Ag @ TiO_2_ NW diffractogram, as well as the rest of the catalysts with other percentages of silver, showed a small shift towards lower angles compared to the pure TiO_2_ NWs. This displacement has been attributed to the incorporation of the silver nanoparticles on the surface of the support [29]. No silver peak was observed at low silver loading (1%Au@TiO_2_ NWs and 3%Au@TiO_2_ NWs). Only 10%Ag@TiO_2_ NWs presented two peaks at ca. 38° (111) and 46° (200), ascribed to the presence of silver nanoparticles with fcc unit cell.

TiO_2_-P25 is a mixture of 70% anatase and 30% rutile. The characteristic peaks of anatase (JCPDS 21-1272) can be found at ca. 25° (101), 38° (004), 48° (220), 54° (105), and 55° (211) [28], while the rutile crystalline phase has its characteristic peaks at ca. 27° (110), 36° (101), 41° (111), and 54° (211) (JCPDS 34-180) [28]. All the peaks corresponding to both anatase and rutile are present in the diffraction pattern of Figure 3B–D. The characteristic peak of rutile (ca. 27°) was not present in any silver loadings. Only the 10%Ag@TiO_2_-P25 showed one peak assigned to the face-centered cubic (fcc) structure of the adsorbed Ag metal nanoparticles corresponding to the h k l parameters (200) at ca. 46° [30].

The XRD patterns of silver-based ZnO NWs and commercial ZnO did not show major differences (Figure 3C,D). Peaks shown at ca. 32° (100), 34.8° (002), 36° (101), 47.5° (102), 56.2° (110), 62.8° (103), 66° (200), 67.5° (112), and 68.8° (201) were ascribed to the ZnO wurtzite phase (JCPDS 396-1451) [31]. The presence of silver nanoparticles was only detected in 10%Ag@ZnO commercial and 10%Ag@ZnO NWs at ca. 38° (111) and not in catalysts with lower silver percentages.

Figure 4 shows the Raman spectra of pristine catalysts and catalysts with 10 wt.% Ag. The anatase phase shows major bands at ca. 150, 395, 515, and 638 cm^−1^ [32] that can be observed in the bare TiO_2_-P25 and 10%Ag@TiO_2_-P25 (Figure 4A, a,b) catalysts, being attributed to the five Raman-active modes of the anatase phase, corresponding to E_g(1)_, B_1g(1)_, A_1g_ + B_1g(2)_, and E_g(2)_ vibrational modes. The typical Raman bands resulting from the rutile phase appear at ca. 143 cm^−1^ (superimposed with the 145 cm^−1^ band from the anatase), 235, 455, and 612 cm^−1^ and have been ascribed to the B_1g_, E_g_, and A_1g_ vibrational modes [32]. In the case of the 10%Ag@TiO_2_-P25 (Figure 4A,b), and in catalysts with lower percentages of silver, the peak corresponding to the presence of silver nanoparticles was not detected. The only change observed by increasing the silver loading of the samples was a slight decrease in the intensity of the characteristic peaks. This effect has been attributed directly to the presence of silver [33]. In TiO_2_ NWs (Figure 4B,c), the characteristic Raman bands of TiO_2_ rutile can be found at ca. 275 and 475 cm^−1^ [34]. As can be seen in Figure 4B,c, these bands are present, confirming that rutile is the only crystalline phase identified in the as-synthesized TiO_2_ NWs. As already shown before for catalysts based on P25, no peaks corresponding to the presence of silver nanoparticles were detected (see Figure 4B,d).

In the case of commercial ZnO and ZnO NWs, no relevant differences were observed (Figure 4C,D). The wurtzite phase shows main bands at ca. 327, 378, 437, and 1050 cm^−1^ [35]. The 1150 cm^−1^ band, observed in both the commercial catalyst (Figure 4C,e) and the as-synthesized ZnO NWs (Figure 4D,g), has been attributed to overtones and/or combination bands [36]. The narrow strong band at 437 cm^−1^ was assigned to the *E_2_* modes, involving mainly Zn motion, corresponding to the band characteristic of the wurtzite phase [34]. The band at 378 cm^−1^ was ascribed to the A_1T_ mode, indicating some degree of structural disorder in the ZnO lattice [36]. No bands associated with Ag NPs were detected in 10%Ag@ZnO commercial (Figure 4C,f), 10%ZnO NWs (Figure 4D,h), or in catalysts with lower percentages of silver.

The two catalyst systems were also characterized by XPS. Figure 5 shows the most relevant transitions of each system, considering the catalysts with the highest silver load (10%Ag@ZnO NWs and 10%Ag@TiO_2_NWs). Figure 5A shows the transitions corresponding to Ti2p of the 10%Ag@TiO_2_NW catalyst. The peaks observed at 464.5 and 458.8 eV have been unambiguously assigned to Ti^4+^, typical of rutile TiO_2_, as identified by XRD [36,37]. In this catalyst, the O1s clearly shows two components at 530.1 and 531.4 eV. The lowest bond energy peak has been assigned to O^2−^ ions in the Ti-O bonds. The shoulder around 531.4 eV has been assigned to O^2−^ ions in the oxygen-deficient regions, as previously referenced in the literature [36,37]. Figure 5C shows the transitions corresponding to Ag3d, with peaks at 374 and 368 eV, and a characteristic spin-orbit splitting of 6.0 eV, which have been clearly assigned to metallic Ag [36,37]. In the case of the 10%Ag@ZnONWs catalyst, the result obtained was also as expected. Figure 5D shows a peak at 1022.1 eV, corresponding to the Zn 2p_3/2_ transition, which has been assigned to Zn^2+^ in ZnO [36,37]. As was the case with the rutile-based catalyst, the peak corresponding to the O1s transition also shows two components: a main component at 530.1 eV, assigned to oxygen in Zn-O bonds, and a shoulder at ca. 531.3 eV that has been assigned to O^2−^ ions in the oxygen-deficient regions [17,38]. In this catalyst, and as can be deduced from the deconvolution carried out (see Figure 5B,E), the ratio corresponding to the shoulder/main peak is lower than in the case of the rutile-based catalyst, which undoubtedly has implications for the catalytic behavior [36,37]. Finally, as with the rutile-based catalyst, the Ag nanoparticles on the surface are found as metallic silver (see Figure 5F).

The pristine catalysts and catalysts with 10 wt.% were also characterized by UV–vis spectroscopy (see Figure 6). The main absorption of TiO_2_ (Figure 6A) was observed at ca. 300 nm, showing low absorption efficiency in the visible range. No additional absorption peaks were detected for any of the Ag@TiO_2_-P25 (Figure 6A,b) and Ag@TiO_2_ NW (Figure 6A,c) catalysts. This result is justified by the limited sensitivity of the spectrophotometer and the small size of the Ag NPs [39].

In the case of ZnO catalysts, the maximum absorption was observed at ca. 360 nm (Figure 6B,d,e,f), with very low absorption efficiency in the visible region. As with TiO_2_-based catalysts, no peaks associated with the presence of Ag NPs were observed.

### 3.2. Photocatalytic Hydrogen Production via Water Splitting

The photocatalytic hydrogen production of the unmodified and silver-based catalysts at wavelengths of 300, 400, and 500 nm is shown in Figure 7. At 320 nm, the highest hydrogen production was obtained with the 10%Ag@ZnO NWs (795 µmol/hg), followed by 10%Ag@TiO_2_ NWs (758 µmol/hg), 10%Ag@TiO_2_-P25 (575 µmol/hg), and 10%Ag@ZnO commercial (483 µmol/hg), respectively. When compared with the production of the unmodified catalysts, all the silver-based catalysts increased their hydrogen production over 420 µmol/hg, with the 10%Ag@TiO_2_ NWs catalyst being the one with the highest difference, 702 µmol/hg. This represented an increase that was 13.5 times greater than bare TiO_2_ NWs. More details can be found in Table 2.

In all cases, the highest amount of hydrogen was obtained with a silver load of 10% by weight, indicating that even higher hydrogen production could be obtained with higher silver loadings. Another important aspect is that the highest hydrogen production was obtained with the catalyst of higher surface area. Different studies [34,40,41,42,43] have found that a key factor in the photocatalytic activity of semiconductors, such as TiO_2_ and ZnO, is their high surface area. A high surface area leads to a higher density of localized states [34], which involve electrons with energies between the conduction and valence bands. These electrons are present due to terminated and unsaturated bonds on the surfaces, providing beneficial charge separation in the form of trapping sites for photo-generated charge carriers [34]. Although the 10%Ag@TiO_2_ NW catalyst has a higher surface area compared to 10%Ag@ZnO NWs (see Table 1), the 10%Ag@ZnO NW catalyst obtained higher hydrogen production. In fact, the amount of hydrogen obtained by the unmodified ZnO NWs was not expected and it is unusual for ZnO. Zhang et al. [43] have reported that one-dimensional nanostructures, such as nanowires, may have greater photocatalytic activity due to their large surface-to-volume ratio, as compared to other morphologies. Additionally, different studies [43,44,45] reveal that surface properties such as surface defects and oxygen vacancies of photocatalysts play a significant role in photocatalytic activity. These studies argue that the crystalline defects of ZnO nanowires exist primarily due to oxygen vacancies and that nanoparticles with crystalline defects can exhibit visible light photocatalysis even without doping with transitional metals.

At 320 nm (UV light), direct photoexcitation of TiO_2_ or ZnO with photons of energy greater than the bandgap (λ < 380 nm) is assumed, leading to the generation of electrons in the semiconductor conduction band and electron holes in the valence band [46]. The electrons in the conduction band will move to the silver nanoparticles, acting as electron buffers and catalytic sites for hydrogen generation [46]. The electron holes will be quenched by the sacrificial electron donors (SO_3_^2−^/SO_4_^2−^) [23,24].

At 400 nm, the highest hydrogen production values were 1065, 963, 648, and 516 µmol/hg, obtained by 10%Ag@TiO_2_ NWs, 10%Ag@ZnO NWs, 10%Ag@TiO_2_-P25, and 10%Ag@ZnO commercial, respectively (Figure 7B). The highest difference in hydrogen production between silver-based and unmodified catalysts was obtained by the 10%Ag@TiO_2_ NWs and bare TiO_2_ NWs (990 µmol/hg). This amount was 14 times higher than that reported by the pristine catalyst. Details of the other maximum amounts and differences can be found in Table 3.

As it was seen at 320 nm, the highest hydrogen production at 400 nm was obtained with the catalysts of 10 wt.% of silver loadings and higher surface areas. This is an indication that it could be possible to increase the amount of hydrogen at higher silver loadings. At 400 nm, the Ag@TiO_2_ NW catalysts produced more hydrogen than the Ag@ZnO NW composites. This could be attributed to the marked difference in surface area between the catalysts (see Table 1) and the band gap energies of the supports. ZnO has a bandgap energy of 3.37 eV, while for TiO_2_, the bandgap is somewhat lower (3.2 eV, anatase); therefore, with a wavelength of 400 nm, it would be easier to promote the excitation of the electrons in the valence band of TiO_2_ than those of ZnO [42].

It is assumed that when irradiated with visible light (λ > 400 nm), photoexcitation of Ag NPs due to plasmon resonance occurs, and electrons from Ag are injected into the TiO_2_ or ZnO conduction band, leading to the generation of holes in the Ag NPs [46]. The water molecule gains the electrons in the conduction band and hydrogen is produced.

At 500 nm, the catalysts with a silver loading of 10% by weight (see Figure 7C) obtained their highest hydrogen production, as seen at other irradiation wavelengths. The highest hydrogen production of the 10%Ag@TiO_2_ NWs, 10%Ag@ZnO NWs, 10%Ag@TiO_2_-P25, and 10%Ag@ZnO commercial was 1119, 921, 653, and 466 µmol/hg, respectively, representing an increase in hydrogen production of ca. 18, 2, 36, and 12 times compared to the bare TiO_2_ NWs, ZnO NWs, TiO_2_-P25, and ZnO commercial catalysts, respectively (see Table 4). Furthermore, as seen at other wavelengths, the larger the surface area, the greater the hydrogen production. This is an indication of the synergism between the deposited Ag NPs and the surface area of the catalysts. As previously mentioned, at wavelengths above 400 nm (λ > 400 nm), the hydrogen production will depend mainly on the Ag NPs and their ability to inject photoexcited electrons into the conduction band of semiconductors.

Table 5 shows recent research on photocatalytic hydrogen production using silver-based catalysts (Ag@TiO_2_ and Ag@ZnO). These investigations consider different parameters such as Ag loading, crystalline structure, morphology of the support and nanoparticles, reaction mixture, and reaction time, among others, when experiments are carried out. Considering the previous scientific literature (see Table 5), the catalysts synthesized in this research gave rise to the highest hydrogen production values achieved by water splitting. However, it is important to note that the experimental parameters used in the literature were not exactly the same as those used in this research.

### 3.3. Photocatalytic Degradation of Ciprofloxacin

The photocatalytic activity of the silver-based catalysts was also studied by the degradation of the antibiotic ciprofloxacin (see Figure 8 and Table 6). The incorporation of Ag NPs increased the percentage of degradation in comparison with the unmodified catalysts. The superior performance of the silver-based catalysts is attributed to the combination of its smaller bandgap and plasmonic effects, which allow visible light energy harvesting and improved charge carrier lifetimes [12]. For the Ag@TiO_2_-P25 system (Figure 8A), the higher degradation was obtained with 5%Ag@TiO_2_-P25. However, it should be noted that with 3%Ag@TiO_2_-P25, the degradation obtained was only ~3% below the maximum, which means that there is no significant difference between 3%Ag@TiO_2_-P25 and 5%Ag@TiO_2_-P25. In addition, 10%Ag@TiO_2_-P25 degraded approximately 70% of the ciprofloxacin within the first 10 min of the reaction. After that time, less than 15% degradation was observed, possibly due to the lack of active sites on the catalyst’s surface [48]. Moreover, 3%Ag@TiO_2_-P25 showed ca. 85% degradation after 40 min of reaction and increased to only 87% at 60 min. According to these results, the water treatment could be stopped at 40 min. After this reaction time, 10% Ag@TiO_2_-P25 did not show a significant increase in degradation, which could indicate that a saturation point was reached.

The Ag@TiO_2_ NW system (Figure 8B) yielded similar results to those mentioned above. The highest degradation percentage, 86%, was obtained with 5%Ag@TiO_2_NWs and showed a continuous degradation pattern, suggesting that there is no saturation point. Similar results have been reported by other authors [19]. Although all the Ag@TiO_2_NWs catalysts have a higher surface area than Ag@TiO_2_-P25, the degradation percentages were better for Ag@TiO_2_-P25 catalysts. Some authors [49] have reported similar results and have justified them as due mainly to the crystalline structure of the catalysts and not so much to the surface area, since it is well known that anatase exhibits better photocatalytic performance than the rutile-type structure. In this regard, TiO_2_-P25 is composed of 70% anatase and 30% rutile, while TiO_2_ NWs are 100% rutile (see Figure 3A).

The Ag@ZnO commercial catalysts (bare ZnO commercial, 1%Ag@ZnO commercial, and 10%Ag@ZnO commercial) showed minor efficiency during the first 10 min of reaction (see Figure 8C). However, after completing the 60 min of reaction, they showed better performance compared to TiO_2_ catalysts. This behavior could be attributed to pH, since TiO_2_ works better in an acidic environment due to the zero-point charge of this material [50]. With 3%Ag@ZnO commercial, ~90% degradation was achieved only after 20 min of reaction. No significant difference was observed between 3%Ag@ZnO commercial and 5%Ag@ZnO commercial catalysts. Moreover, these samples degrade +60% of the ciprofloxacin in the first 10 min of reaction, while the bare ZnO took 30 min to reach this point, clearly indicating that the best performance is achieved when the material is doped with Ag.

Ag@ZnO NW (Figure 8D) catalysts showed lower efficiency compared to the commercial catalyst. All catalysts showed a constant increase over time, suggesting that the catalyst still has active sites available, even after 60 min of reaction, and no saturation point is expected to occur. The highest degradation percentage (ca. 78%) was obtained with 3%Ag@ZnO NWs.

In the case of ZnO commercial and ZnO NWs, both exhibited the same wurtzite crystalline phase (see Figure 3C,D) but demonstrated a significant difference in degradation percentages. These differences are attributed to the agglomeration of particles; the ZnO commercial consists of a finer powder compared to the as-synthesized material. This might cause agglomeration of the as-synthesized catalyst, enhancing the catalyst–catalyst contact instead of the catalyst–antibiotic contact, reducing the degradation efficiency of the system.

To further study the photocatalytic activity of the silver-based catalysts, control experiments and stability tests were conducted for the two catalysts with the highest percentage of degradation (5%Ag@TiO_2_-P25, 3%Ag@ZnO commercial) (Figure 9). The control experiments for both catalysts were similar (see Figure 9A,B). After 60 min of reaction, it was observed that ciprofloxacin is highly recalcitrant and stable under normal or environmental conditions. This is evidenced by the fact that no degradation or significant changes in ciprofloxacin concentration are observed over time without the catalyst or radiation source. From the control experiments, it is shown that photocatalysis is the primary degradation route; this means that neither catalysis nor photolysis are enough to degrade the contaminant. Instead, degradation occurs due to a synergistic effect between the catalyst and the radiation source.

The stability tests (Figure 9C,D) for both catalysts were also very similar. The results showed that even after 60 min of reaction, the catalysts were demonstrated to be highly stable and that no significant chemical change occurred that endangered the efficiency of the materials. After seven cycles, the degradation efficiency decreased by ~7%, which can be estimated as 1% per cycle of use. This rough estimate suggests that these catalysts could be employed for several more cycles without the need for resynthesizing. This means that these materials could decrease the environmental remediation costs if used in contaminated settings.

The results obtained have made it possible to establish the possible mechanisms involved in the two processes studied in this research. Figure 10 shows a possible mechanism for the production of hydrogen via water splitting using Ag@TiO_2_ or Ag@ZnO catalysts under visible and ultraviolet light. When irradiated with ultraviolet light (Figure 10A), direct photoexcitation of TiO_2_ or ZnO with photons with energy larger than the bandgap (λ < 380 nm) is assumed, leading to the generation of electrons in the semiconductor conduction band and electron holes in the valence band [46]. The electron in the conduction band will move to the silver nanoparticles, acting as electron buffers and catalytic sites for hydrogen generation [11,12]. The electron holes will be quenched by the sacrificial electron donors (SO_3_^2−^/SO_4_^2−^). When irradiated with visible light (λ > 400 nm) (Figure 10B), photoexcitation of Ag NPs occurs and electrons from the Ag are injected into the TiO_2_ or ZnO conduction band, leading to the generation of holes in the Ag NPs and electrons in the TiO_2_ or ZnO conduction band [11,12,46]. The water molecule gains the electrons in the conduction band and hydrogen is produced. Holes in the Ag NPs will be quenched by the sacrificial electron donors (SO_3_^2−^/SO_4_^2−^). This proposed mechanism is an oversimplification since different studies [11,12,13] have determined that, due to the silver/titania or silver/zinc oxide interfacial contact, the conduction band of the semiconductor undergoes a shift toward more negative potentials. Thus, the charge distribution between the Ag NPs and the semiconductor causes a shift in the Fermi level toward more negative potentials [13].

The mechanism for the degradation of ciprofloxacin using the silver-based catalysts is shown in Figure 11. Under ultraviolet light (Figure 11A), electrons from the valence band of TiO_2_ or ZnO migrate to the conduction band of the semiconductors and holes in the valence band are formed. The electrons in the conduction band move to the Ag NPs, acting as an electron buffer, and these electrons may react with adsorbed molecular oxygen to generate superoxide anions, which in turn can react with water molecules to form hydroxyl radicals. These free radicals are very efficient in the photodegradation of organic pollutants such as ciprofloxacin [12]. The holes formed in the valence band of the semiconductors then will promote the oxidation of ciprofloxacin and therefore lead to degradation. When irradiated with visible light (Figure 11B), Ag NPs form an interface with semiconducting materials and a Schottky barrier is formed, resulting in a new Fermi level and a high number of electrons due to the presence of metallic Ag [12]. Additionally, free electrons are stimulated through the silver surface plasmon resonance mechanism and can move into the conduction band of the partially reduced TiO_2_ or ZnO [12]. These conduction band electrons may react with adsorbed molecular oxygen to generate superoxide anions, then react with water molecules and form hydroxyl radicals that will eventually oxidize the ciprofloxacin. In future research, already under development, the different intermediate compounds generated during the ciprofloxacin degradation process will be analyzed, which will allow us to establish the catalytic photodegradation mechanisms that, eventually, will allow the process to be improved.

## 4. Conclusions

Silver-based catalysts were synthesized to study their ability to mimic natural processes such as the splitting of water and the degradation of pollutants. The photocatalytic activity of the composites was studied by the production of hydrogen via water splitting using UV–vis light and the degradation of the antibiotic ciprofloxacin.

The catalyst with the highest hydrogen production was 10%Ag@TiO_2_ NW (1119 µmol/hg), being 18 times greater than the amount obtained with the pristine TiO_2_ NW catalyst. The most dramatic difference in hydrogen production was obtained with 10%Ag@TiO_2_-P25 (635 µmol/hg), being 36 times greater than the amount reported for the unmodified TiO_2_-P25 (18 µmol/hg). The enhancement of the catalytic activity is attributed to a synergism between the silver nanoparticles incorporated and the high surface area of the composites.

In the case of the degradation of ciprofloxacin, all the silver-based catalysts degraded more than 70% of the antibiotic in 60 min. The catalyst that exhibited the best result was 3%Ag@ZnO commercial with 99.72% of degradation. The control experiments and stability tests showed that photocatalysis was the route of degradation and the selected silver-based catalysts were stable after seven cycles, with less than 1% loss of efficiency per cycle. These results suggest that the catalysts could be employed in several more cycles without the need for resynthesizing, thus reducing remediation costs.

The results obtained in this research introduce an immense field of possibilities for the development of high-performance photocatalysts. The systems studied have proven to be very active in the production of hydrogen and in the degradation of ciprofloxacin, so perhaps it is towards the development of multi-application systems that we will have to redouble our efforts in the future.

## Figures and Tables

**Figure 1 biomimetics-06-00004-f001:**
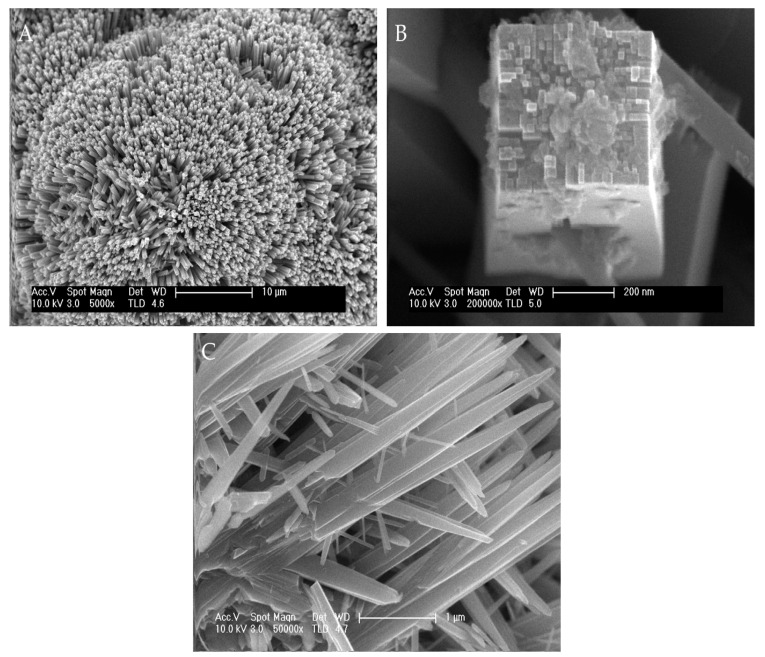
Field emission scanning electron microscopy (FE-SEM) images of the as-synthesized TiO_2_ Nanowires (NWs) at 5000× (**A**), 20,000× (**B**), and ZnO NWs at 5000× (**C**).

**Figure 2 biomimetics-06-00004-f002:**
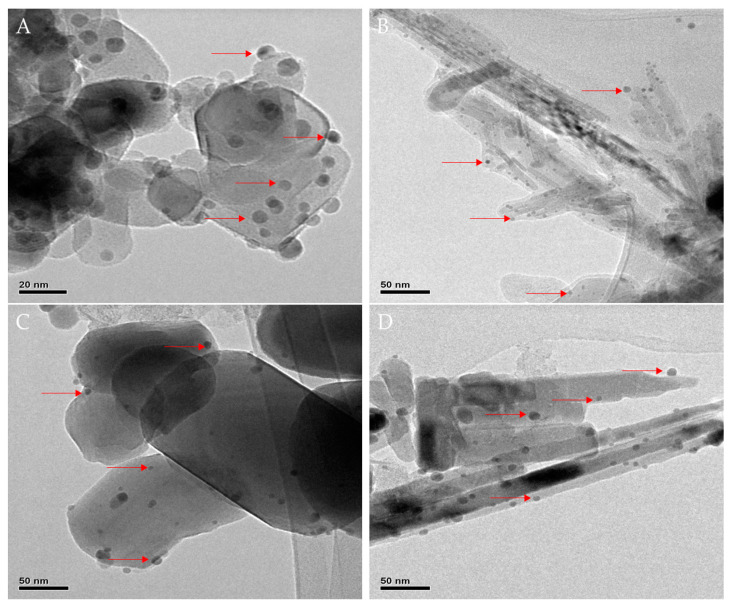
High-resolution transmittance electron microscopy (HRTEM) images of 10%Ag@TiO_2_-P25 (**A**), 10%Ag@TiO_2_ NWs (**B**), 10%Ag@ZnO commercial (**C**), and 10%Ag@ZnO NWs (**D**) catalysts. The red arrows indicate the presence of Ag Nanoparticles (NPs).

**Figure 3 biomimetics-06-00004-f003:**
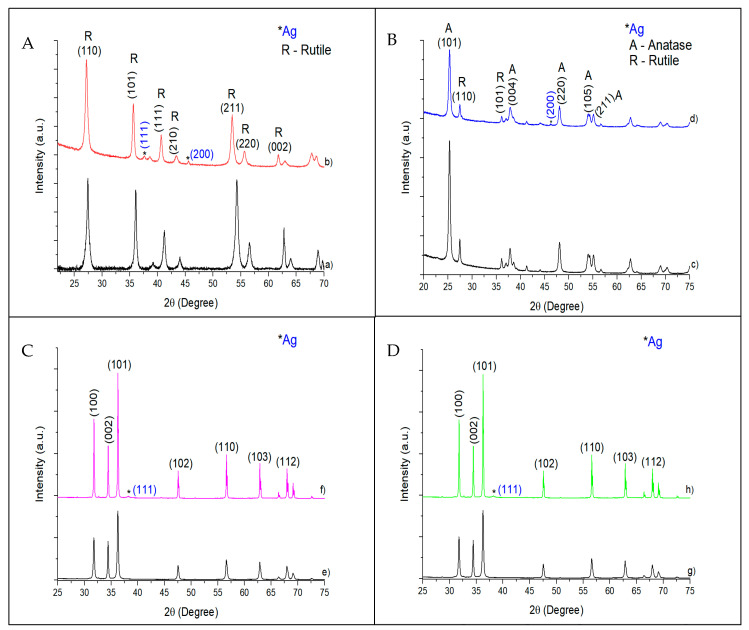
X-ray diffraction patterns (XRD) of TiO_2_ NW (**A**,a), 10%Ag@TiO_2_ NW (**A**,b), TiO_2_-P25 (**B**,c), 10%Ag@TiO_2_-P25 (**B**,d), ZnO commercial (**C**,e), 10%Ag@ZnO commercial (**C**,f), ZnO NW (**D**,g), and 10%Ag@ZnO NW (**D**,h) catalysts.

**Figure 4 biomimetics-06-00004-f004:**
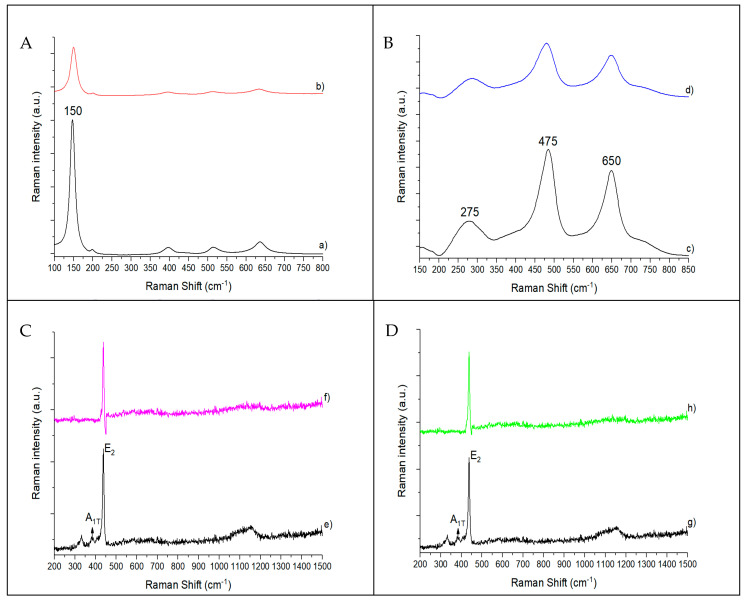
Raman spectra of TiO_2_-P25 (**A**,a), 10%Ag@TiO_2_-P25 (**A**,b), TiO_2_ NWs (**B**,c), 10%Ag@TiO_2_ NWs (**B**,d), ZnO commercial (**C**,e), 10%Ag@ZnO commercial (**C**,f), ZnO NWs (**D**,g), and 10%Ag@ZnO NWs (**D**,h) catalysts.

**Figure 5 biomimetics-06-00004-f005:**
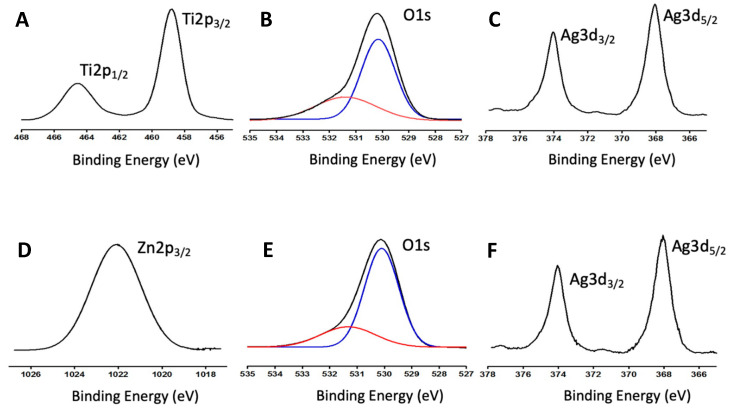
X-ray photoelectron spectroscopy (XPS) spectra of Ti 2p_1/2_, O 1s, Ag3d_3/2_, Ag3d_5/2_ taken from the 10%Ag@TiO_2_ NW catalyst (**A**–**C**), and Zn 2p_1/2_, O 1s, Ag3d_3/2_, and Ag3d_5/2_ taken from the 10%ZnO NW composite (**D**–**F**).

**Figure 6 biomimetics-06-00004-f006:**
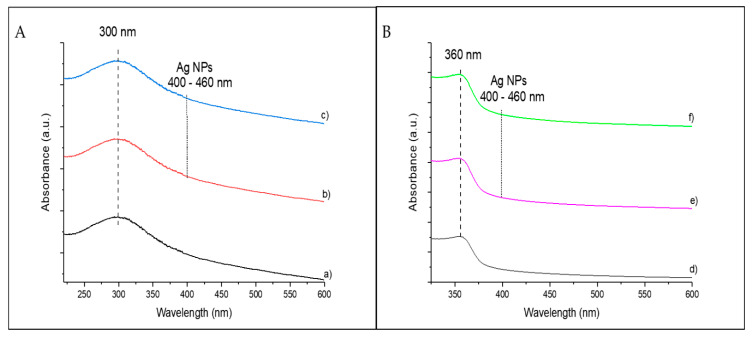
UV–vis spectra of TiO_2_ NW (**A**,a), 10%Ag@TiO_2_-P25 (**A**,b), 10%Ag@TiO_2_ NW (**A**,c), ZnO NW (**B**,d), 10%Ag@ZnO commercial (**B**,e), and 10%Ag@ZnO NW (**B**,f) catalysts. The region in which the presence of Ag NPs should be observed is indicated.

**Figure 7 biomimetics-06-00004-f007:**
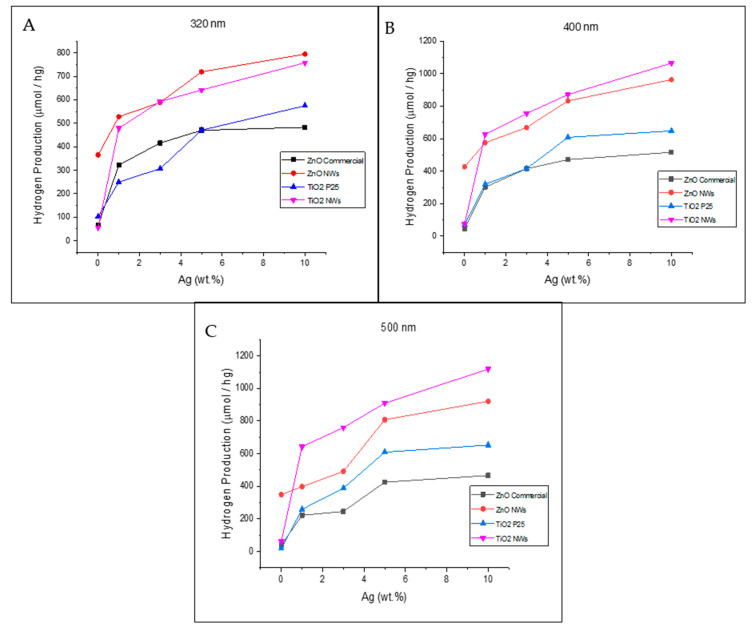
Photocatalytic hydrogen production of the different silver-based catalysts under irradiation at 320 (**A**), 400 (**B**), and 500 nm (**C**).

**Figure 8 biomimetics-06-00004-f008:**
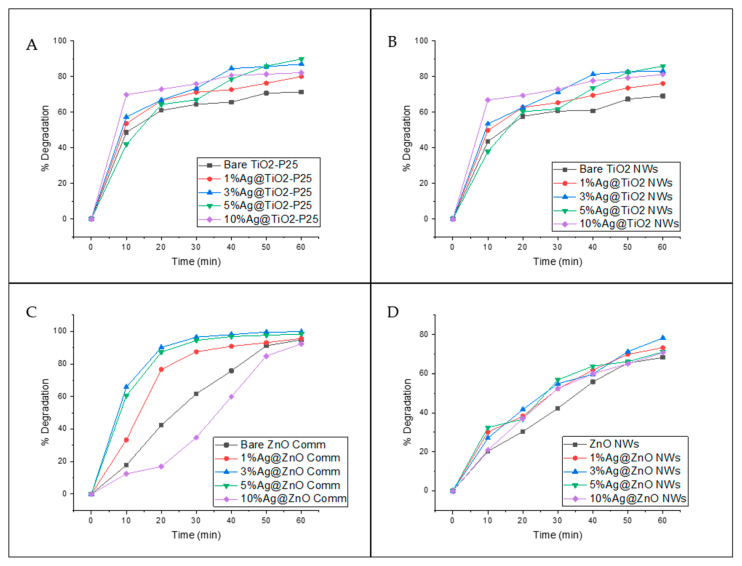
Percentage of degradation of ciprofloxacin using Ag@TiO_2_-P25 (**A**), Ag@TiO_2_ NW (**B**), Ag@ZnO commercial (**C**), and Ag@ZnO NW (**D**) catalysts.

**Figure 9 biomimetics-06-00004-f009:**
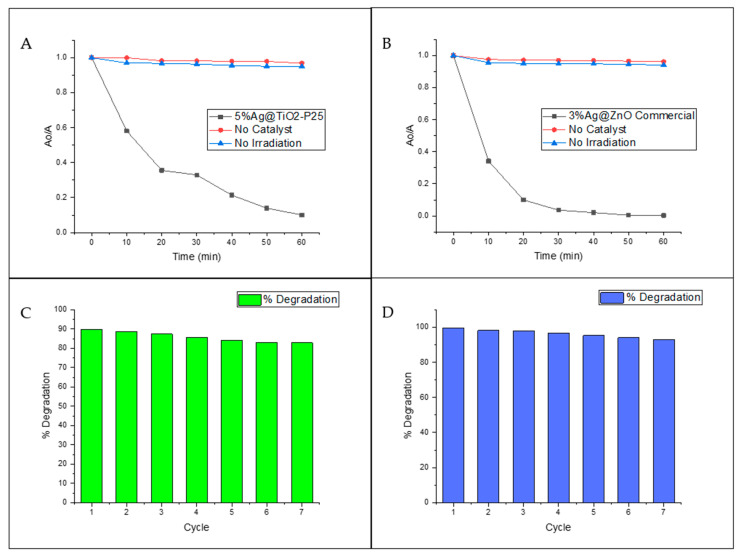
Control experiments and stability tests for the 5%Ag@TiO_2_-P25 (**A**,**C**) and 3%Ag@ZnO commercial (**B**,**D**) catalysts.

**Figure 10 biomimetics-06-00004-f010:**
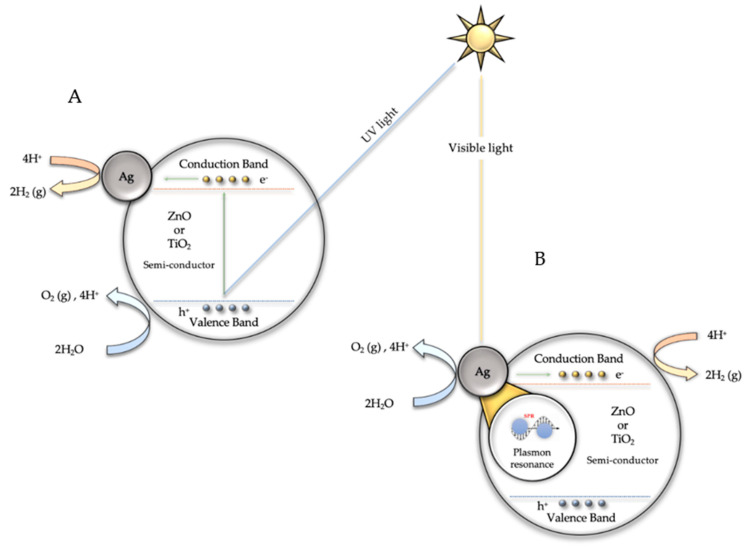
Mechanism for the production of hydrogen via water splitting using Ag@TiO_2_ or Ag@ZnO catalysts under ultraviolet (**A**) and visible light (**B**).

**Figure 11 biomimetics-06-00004-f011:**
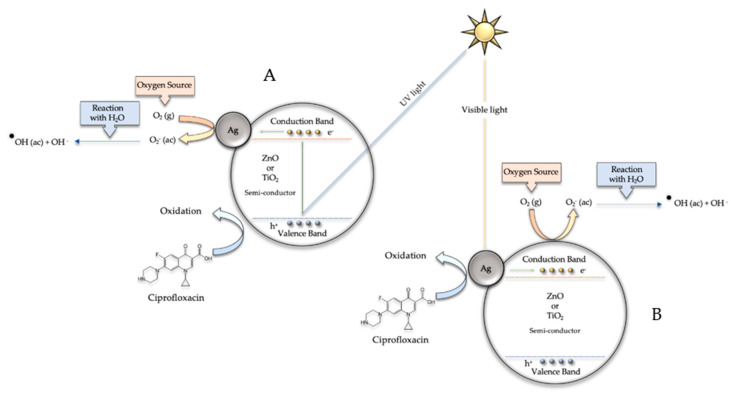
Mechanism for the degradation of ciprofloxacin using the silver-based catalysts under ultraviolet (**A**) and visible light (**B**).

**Table 1 biomimetics-06-00004-t001:** Brunauer, Emmett and Teller (BET) surface area of the silver-based catalysts.

	TiO_2_-P25 (m^2^·g^−^^1^)	ZnO Commercial (m^2^·g^−^^1^)	TiO_2_ NWs (m^2^·g^−^^1^)	ZnO NWs (m^2^·g^−^^1^)
0% (Bare)	53	18	403	160
1%Ag	63	78	417	167
3%Ag	72	81	426	179
5%Ag	81	99	438	194
10%Ag	89	112	443	202

**Table 2 biomimetics-06-00004-t002:** Silver-based catalysts with the highest hydrogen production and their difference from the unmodified supports at a wavelength of 320 nm *.

Catalyst	Maximum H_2_ Production; Silver-Based Catalysts (µmol/gh)	Maximum H_2_ Production; Unmodified Catalyst (µmol/gh)	Difference in H_2_ Production (µmol/gh)	Times Greater Than Unmodified Catalyst
10%AgTiO_2_ NWs	758	TiO_2_ NWs; 56	702	13.53
10%AgTiO_2_-P25	575	TiO_2_-P25; 103	389	5.58
10%AgZnO NWs	795	ZnO NWs; 365	430	2.18
10%AgZnO comm **	483	ZnO Comm * 66	417	7.32

* Only the highest hydrogen production from all the silver-based catalysts is shown, ** Commercial.

**Table 3 biomimetics-06-00004-t003:** Silver-based catalysts with the highest hydrogen production and their difference from the unmodified supports at a wavelength of 400 nm *.

Catalyst	Maximum H_2_ Production; Silver-Based Catalysts (µmol/gh)	Maximum H_2_ Production Unmodified Catalyst (µmol/gh)	Difference in H_2_ Production (µmol/gh)	Times Greater Than Unmodified Catalyst
10%AgTiO_2_ NWs	1065	TiO_2_ NWs; 75	990	14.20
10%AgTiO_2_-P25	648	TiO_2_-P25; 71	577	9.13
10%AgZnO NWs	963	ZnO NWs; 427	430	2.26
10%AgZnO comm. **	516	ZnO Comm * 45	471	11.47

* Only the highest hydrogen production from all the silver-based catalysts is shown, ** Commercial.

**Table 4 biomimetics-06-00004-t004:** Silver-based catalysts with the highest hydrogen production and their difference from the unmodified supports at a wavelength of 500 nm *.

Catalyst	Maximum H_2_ Production Silver-Based Catalysts (µmol/gh)	Maximum H_2_ Production Unmodified Catalyst (µmol/gh)	Difference in H_2_ Production (µmol/gh)	Times Greater Than Unmodified Catalyst
10%AgTiO_2_ NWs	1119	TiO_2_ NWs; 62	1057	18.05
10%AgTiO_2_-P25	653	TiO_2_-P25; 18	635	36.28
10%AgZnO NWs	921	ZnO NWs; 349	572	2.64
10%AgZnO comm. **	466	ZnO Comm *; 38	428	12.26

* Only the highest hydrogen production from all the silver-based catalysts is shown, ** Commercial.

**Table 5 biomimetics-06-00004-t005:** Recent works for photocatalytic hydrogen production using Ag@TiO_2_ and Ag@ZnO catalysts.

Reference	H_2_ Production (µmol)	Source (nm)	Irradiation Time (h)	TiO_2_ or ZnO Crystal Structure *	Reaction Mixture	Ag (wt.%)
[12]	910	λ > 400	2	TiO_2_; A	Water: Methanol	14
[47]	90	λ = 457	8	TiO_2_; A	Water: 0.1 M Na_2_S	2
[15]	443.6	λ > 400	4	ZnO; W	Water: 0.25 M Na_2_S, 0.35 M Na_2_SO_3_	3.12
[16]	55	λ > 400	8	ZnO; W	Water: 3.0 g Na_2_S, 2.2 g Na_2_SO_3_	0.5
This work	1119	λ = 500	2	TiO_2_ NWs; R	Water: 0.5 M Na_2_S, 0.03 M Na_2_SO_3_	10
This work	653	λ = 500	2	TiO_2_-P25; A, R	Water: 0.5 M Na_2_S, 0.03 M Na_2_SO_3_	10
This work	963	λ = 400	2	ZnO NWs; W	Water: 0.5 M Na_2_S, 0.03 M Na_2_SO_3_	10
This work	516	λ = 400	2	ZnO comm. ** W	Water: 0.5 M Na_2_S, 0.03 M Na_2_SO_3_	10

* A = anatase, R = rutile, W = wurtzite; ** commercial.

**Table 6 biomimetics-06-00004-t006:** Degradation percentages of all silver-based catalysts after 60 min of reaction.

Ag Loading (wt.%)	TiO_2_ NWs (%)	TiO_2_-P25 (%)	ZnO NWs (%)	ZnO Commercial (%)
0	69.12	71.36	68.26	94.83
1	76.11	80.11	73.37	95.60
3	83.09	87.09	78.26	99.72
5	85.92	89.92	71.11	98.39
10	81.26	82.26	70.65	92.31

## Data Availability

Not applicable.

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
