# Peer review of "Photocatalytic Activity of Silver-Based Biomimetics Composites"

_biomimetics, 2021, doi:10.3390/biomimetics6010004_

Round 1
Reviewer 1 Report
This manuscript presents the synthesis methodology of the Ag@ZnO and Ag@TiO2 NWs for photocatalytic water splitting and photodegration of ciprofloxacin. The manuscript is coherent and shows complete material characterization. This manuscript is suitable for publication after the integration of the following remarks.
- Water splitting was written in the manuscript, but no sacrificial agent was used to screen the generation of O2.
- Please , note what the NWs stand for when it is first mentioned.
- “Before the silver deposition, the average surface area of the as- synthesized TiO2 NWs and ZnO NWs were 403 m2g-1 and 160 m2g-1, respectively, whereas the surface area of the commercial ones (TiO2-P25 and ZnO) were 53 m2g-1 and 18 m2g-1, respectively. These results clearly show that the synthesis processes used to obtain TiO2 NWs and ZnO NWs allowed to obtain high-surface supports. Unexpectedly, the addition of silver nanoparticles on the pristine catalysts resulted in an increase in surface area. This effect is much more significant in composites with the highest percentage of silver on the surface, showing increases of about 36 m2g-1 (10%Ag@TiO2-P25), 94 m2g-1 (10%Ag@Zno Commercial), 40 m2g-1 (10%Ag@TiO2 NWs), and 42 m2g-1 (10%Ag@ZnO NWs), respectively. “ Here the authors demonstrate that the surface area is increasing with the modification of Ag. However, it is misleading to mention increased numbers instead of stating the surface area that has been obtained.
Author Response
Response to reviewer 1
(1) Reviewer-1's comment: “Water splitting was written in the manuscript, but no sacrificial agent was used to screen the generation of O2”
The interest of synthesized nanomaterials is based on their potential application for the generation of hydrogen and their proven use for the photodegradation of polluting compounds, such as ciprofloxacin.
For catalytic water splitting, the addition of a co-catalyst on the catalyst surface (in our case, Ag), and the presence of different sacrificial reagents in the reaction mixture, are required. These components favor the photocatalytic activity and increase the gas (H2 and O2) production, due to their action as electron or hole scavenger, respectively [1, 2]. To improve the clarity of our research, section 2.5 of the revised version of the manuscript has been modified.
1 Jafari T, Moharreri E, Amin AS, Miao R, Song W, Suib SL. Photocatalytic Water Splitting-The Untamed Dream: A Review of Recent Advances. Molecules 2016, 21, 900.
2 Clarizia L, Spasiano D, Di Somma I, Marotta R, Andreozzi R, Dionysiou DD. Copper modified-TiO2 catalysts for hydrogen generation through photoreforming of organics. A short review. Int. J. Hydrogen Energy 2014, 39, 16812.
(2) Reviewer-1's comment: “Please, note what the NWs stand for when it is first mentioned.”
The definition of NW has been introduced in the Abstract of the revised manuscript.
(3) Reviewer-1's comment: ““Before the silver deposition, the average surface area of the as- synthesized TiO2 NWs and ZnO NWs were 403 m2g-1 and 160 m2g-1, respectively, whereas the surface area of the commercial ones (TiO2-P25 and ZnO) were 53 m2g-1 and 18 m2g-1, respectively. These results clearly show that the synthesis processes used to obtain TiO2 NWs and ZnO NWs allowed to obtain high-surface supports. Unexpectedly, the addition of silver nanoparticles on the pristine catalysts resulted in an increase in surface area. This effect is much more significant in composites with the highest percentage of silver on the surface, showing increases of about 36 m2g-1 (10%Ag@TiO2-P25), 94 m2g-1 (10%Ag@ZnO Commercial), 40 m2g-1 (10%Ag@TiO2 NWs), and 42 m2g-1 (10%Ag@ZnO NWs), respectively.” Here the authors demonstrate that the surface area is increasing with the modification of Ag. However, it is misleading to mention increased numbers instead of stating the surface area that has been obtained.”
The BET surface area values reported in the investigation are correct, as are the area increases as the silver loading increases. However, and in response to the reviewer's observation, we have eliminated any allusion to area variations in the revised version of the manuscript.

Reviewer 2 Report
This article presents the results of the authors' study of photocatalytic properties of silver containing TiO2 and ZnO surfaces. The results are well presented and the experiments well performed. However, there is no novel or ground breaking aspects to the work presented and as such, it appreas more like a technical report. Further analysis to support the proposed mechanisms for photocatalytic activities would be beneficial, as well as evidence-supported explanation of the reason for reduced reaction rates for ciprofloxacin with time.
Author Response
Response to reviewer 2
(1) Reviewer-2's comment: “This article presents the results of the authors' study of photocatalytic properties of silver containing TiO2 and ZnO surfaces. The results are well presented, and the experiments well performed. However, there is no novel or ground breaking aspects to the work presented and as such, it appears more like a technical report. Further analysis to support the proposed mechanisms for photocatalytic activities would be beneficial, as well as evidence-supported explanation of the reason for reduced reaction rates for ciprofloxacin with time.
The mechanisms proposed in this investigation, both for the water splitting process and for the photodegradation of ciprofloxacin (represented in Figures 10 and 11 of the manuscript), have been widely proposed through previous works published on similar systems. We consider that these catalytic mechanisms are sufficiently robust, although they will need to be tested in future research. These future investigations require the use of different metallic cocatalysts (silver, gold, platinum, etc.) and of different sacrificial reagents to control the electron or hole scavenger pathways, which will allow to establish the precise behavior mechanism of these catalysts. These investigations, due to their complexity, deviate enormously from the objectives proposed in the present investigation, although we hope that they can be addressed in the future.

Round 2
Reviewer 2 Report
Unfortunately the authors have not addressed my original concern about the novelty and have only confirmed my concerns by stating that the mechanisms proposed in their study are widely proposed through previous works published on similar systems. Considering this and the extent of available publications on such systems, I am still unconvinced that this manuscript adds to the existing knowledge.